# Mind the Gap: A Causal Perspective on Bias Amplification in Prediction & Decision-Making

**Drago Plecko** and **Elias Bareinboim**

Causal Artificial Intelligence Lab
Columbia University
dp3144@columbia.edu, eb@cs.columbia.edu

## Abstract

As society increasingly relies on AI-based tools for decision-making in socially sensitive domains, investigating fairness and equity of such automated systems has become a critical field of inquiry. Most of the literature in fair machine learning focuses on defining and achieving fairness criteria in the context of prediction, while not explicitly focusing on how these predictions may be used later on in the pipeline. For instance, if commonly used criteria, such as independence or sufficiency, are satisfied for a prediction score $S$ used for binary classification, they need not be satisfied after an application of a simple thresholding operation on $S$ (as commonly used in practice). In this paper, we take an important step to address this issue in numerous statistical and causal notions of fairness. We introduce the notion of a margin complement, which measures how much a prediction score $S$ changes due to a thresholding operation. We then demonstrate that the marginal difference in the optimal 0/1 predictor $\widehat{Y}$ between groups, written $P(\hat{y} \mid x_1) - P(\hat{y} \mid x_0)$, can be causally decomposed into the influences of $X$ on the $L_2$-optimal prediction score $S$ and the influences of $X$ on the margin complement $M$, along different causal pathways (direct, indirect, spurious). We then show that under suitable causal assumptions, the influences of $X$ on the prediction score $S$ are equal to the influences of $X$ on the true outcome $Y$. This yields a new decomposition of the disparity in the predictor $\widehat{Y}$ that allows us to disentangle causal differences inherited from the true outcome $Y$ that exists in the real world vs. those coming from the optimization procedure itself. This observation highlights the need for more regulatory oversight due to the potential for bias amplification, and to address this issue we introduce new notions of *weak* and *strong* business necessity, together with an algorithm for assessing whether these notions are satisfied. We apply our method to three real-world datasets and derive new insights on bias amplification in prediction and decision-making.

## 1 Introduction

Automated systems based on machine learning and artificial intelligence are increasingly used for decision-making in a variety of real-world settings. These applications include hiring decisions, university admissions, law enforcement, credit lending and loan approvals, health care interventions, and many other high-stakes scenarios in which the automated system may significantly affect the well-being of individuals [Khandani et al., 2010, Mahoney and Mohen, 2007, Brennan et al., 2009]. In this context, society is increasingly concerned about the implications and consequences of using automated systems, compared to the currently implemented decision processes. Prior works highlight the potential of automated systems to perpetuate or even amplify inequities between demographic groups, with a range of examples from decision support systems for (among others) sentencing

38th Conference on Neural Information Processing Systems (NeurIPS 2024).

Angwin et al. [2016], face-detection Buolamwini and Gebru [2018], online advertising Sweeney [2013], Datta et al. [2015], and authentication Sanburn [2015]. Notably, issues of unfairness and discrimination are also pervasive in settings in which decisions are made by humans. Some well-studied examples include the gender pay gap, supported by a decades-long literature [Blau and Kahn, 1992, 2017], or the racial bias in criminal sentencing [Sweeney and Haney, 1992, Pager, 2003], just to cite a few. Therefore, AI systems designed to make decisions may often be trained with data that contains various historical biases and past discriminatory decisions against certain protected groups, constituting a large part of the underlying problem. In this work, we specifically focus on investigating when automated systems may potentially lead to an even more discriminatory process, possibly amplifying already existing differences between groups.

Within this context, it is useful to distinguish between different tasks appearing in the growing literature on fair machine learning. One can distinguish three specific and different tasks, namely (1) bias detection and quantification for exisiting outcomes or decision policies; (2) construction of fair predictions of an outcome; (3) construction of fair decision-making policies that are intended to be implemented in the real-world. Interestingly, a large portion of the literature in fair ML focuses on the second task of fair prediction, and what is often left unaddressed is how these predictions may be used later on in the pipeline, and what kind of consequences they may have. For instance, consider a prediction score $S$ for a binary outcome $Y$ that satisfies well-known fairness criteria, such as independence (demographic parity [Darlington, 1971]) or sufficiency (calibration [Chouldechova, 2017]). After a simple thresholding operation, commonly applied in settings with a binary outcome, the resulting predictor is no longer guaranteed to satisfy independence or sufficiency, and the previously provided fairness guarantees may be entirely lost. The same behavior can be observed for numerous other measures.

These difficulties do not apply only to statistical measures of fairness. Recently, a growing literature has explored causal approaches to fair machine learning [Kusner et al., 2017, Kilbertus et al., 2017, Nabi and Shpitser, 2018, Zhang and Bareinboim, 2018b,a, Wu et al., 2019, Chiappa, 2019, Plečko and Meinshausen, 2020, Plečko and Bareinboim, 2024], which have two major benefits. First, they allow for human-understandable and interpretable definitions and metrics of fairness, which are tied to the causal mechanisms transmitting the change between groups. Secondly, they offer a language that is aligned with the legal notions of discrimination, such as the disparate impact doctrine. In particular, causal approaches allow for considerations of business necessity – which aim to ellucidate which covariates may be justifiably used by decision-makers even if their usage implies a disparity between groups. However, causal approaches to fairness also suffer from the above-discussed issues – namely, a guarantee of absence of a causal influence from the protected attribute $X$ onto a predictor $S$ need not hold true after the predictor is thresholded [Plečko and Bareinboim, 2024]. Therefore, within the causal approach, there is also a major need for a better understanding of how probabilistic predictions are translated into binary predictions or decisions.

In this work, we take an important step in the direction of addressing this issue. We work in a setting with a binary label $Y$, and the goal is to provide a binary prediction $\widehat{Y}$ or a binary decision $D$. Our approach is particularly suitable for settings in which the utility of the decision is monotonic with respect to the conditional probability of $Y$ being positive, written $P(Y \mid \text{covariates})$, but the developed tools also have ramifications for more general utilities. Examples that fall under our scope are numerous; for instance, the utility of admitting a student to the university ($D$) is often monotonic in the probability that the student successfully graduates ($Y$). In the context of criminal justice, decisions of detention ($D$) are used to prevent recidivism, and the utility of the decision is monotonic in the probability that the individual recidivates ($Y$). Finally, various preventive measures in healthcare ($D$, such as vaccination, screening tests, etc.) are considered to be best applied to individuals with the highest risk of developing a target disease or suffering a negative outcome ($Y$). We now provide an illustrative two-variable example for one of the key insights of our paper:

**Example 1** (Disparities in Hiring). *Consider a company deciding to hire employees using an automated system for the first time. From a previous hiring cycle when humans were in charge, the company has access to data on gender $X$ ($x_0$ for female, $x_1$ for male) and the hiring outcome $Y$ (1 for being hired, 0 otherwise). The true underlying mechanisms of the system are given by:*

$$X \leftarrow \mathbb{1}(U_X < 0.5) \tag{1}$$

$$Y \leftarrow \begin{cases} \mathbb{1}(U_Y < p_0) \text{ if } X = x_0 \\ \mathbb{1}(U_Y < p_1) \text{ if } X = x_1 \end{cases} \tag{2}$$

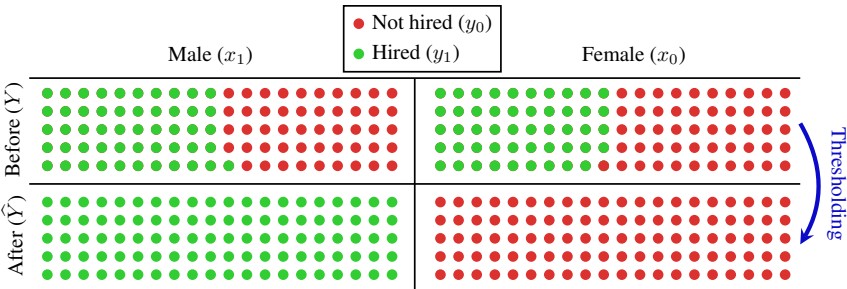

Figure 1: Visualization of hiring disparities from Ex. 1.

*where $U_X, U_Y \sim Unif[0, 1]$. In words, an applicant is female with a 50% probability, and the probability of being hired as a female $x_0$ is $p_0$, whereas for males $x_1$ the probability is $p_1$. The company finds the optimal prediction score $S$ to be $S(x) = p_x$. The optimal 0/1 predictor $\widehat{Y}$, which will also be the company's decision, is given by $\widehat{Y}(x) = \mathbb{1}(S(x) \geq \frac{1}{2})$, meaning that the company will threshold the predictions at $\frac{1}{2}$. Suppose that $p_0 = 0.49, p_1 = 0.51$, and consider the visualization in Fig. 1. The probability $p_0 = 0.49$ means that 49 out of 100 females were hired, while 51 were not. After thresholding, $\widehat{Y}(x_0) = \mathbb{1}(p_0 \geq \frac{1}{2}) = 0$ for each female applicant, meaning that the thresholding operation maps the prediction of each individual to 0, even though 49/100 would have a positive outcome (Fig. 1 right). Similarly, for $p_1 = 0.51$, we have that 51/100 male applicants would be hired, resulting in a thresholded predictor $\widehat{Y}(x_1) = 1$ for all the applicants, even though 49/100 would not have a positive outcome (Fig. 1 left). Therefore, the gender disparity in hiring after introducing the automated predictor $\widehat{Y}$ is 100%, compared to a 2% disparity in the outcome $Y$ before introducing $\widehat{Y}$. Formally, the disparity in the optimal 0/1 predictor $P(\widehat{y} \mid x_1) - P(\widehat{y} \mid x_0) = \mathbb{1}(p_1 \geq \frac{1}{2}) - \mathbb{1}(p_0 \geq \frac{1}{2})$ can be decomposed as:*

$$P(\widehat{y} \mid x_1) - P(\widehat{y} \mid x_0) = \underbrace{p_1 - p_0}_{Term\ I} + \underbrace{(\mathbb{1}(p_1 \geq \frac{1}{2}) - p1) - (\mathbb{1}(p_0 \geq \frac{1}{2}) - p_0)}_{Term\ II}. \qquad (3)$$

*Term I measures the disparity coming from the true outcome $Y$ (2% disparity), which is equal to the disparity in the prediction score $S$, written $P(s \mid x_1) - P(s \mid x_0)$. Term II measures the contribution coming from thresholding the prediction score to obtain an optimal 0/1 prediction (98% disparity).* □

The above example illustrates a canonical point in a simple setting: a small disparity in the outcome $Y$, and consequently the prediction score $S$, may result in a large disparity in the optimal 0/1 predictor $\widehat{Y}$, a case we call *bias amplification*. Contrary to this, a large disparity in $S$ may also result in a small disparity in $\widehat{Y}$, a case we call *bias amelioration*.

In the remainder of the manuscript, our goal is to provide a decomposition of the disparity in a thresholded predictor $\widehat{Y}$ into the disparity in true outcome $Y$ and the disparity originating from optimization procedure, but *along each causal pathway* between the protected attribute $X$ and the predictor $\widehat{Y}$. In particular, our contributions are the following:

(1) We introduce the notion of margin complement (Def. 1), and provide a path-specific decomposition of the disparity in the 0/1 predictor $\widehat{Y}$ into its contributions from the optimal score predictor $S$ and the margin complement $M$ (Thm. 1),

(2) We prove that under suitable assumptions, the causal decomposition of the optimal prediction score $S$ is equivalent with the causal decomposition of the true outcome $Y$ (Thm. 2). This allows us to obtain a new decomposition of the disparity in $\widehat{Y}$ into contributions from $Y$ and the margin complement $M$ (Cor. 3),

(3) Motivated by the above decompositions, we introduce a new concept of weak and strong business necessity (Def. 3), highlighting a new need for regulatory instructions in the context of automated systems. We provide an algorithm for assessing fairness under considerations of weak and strong business necessity (Alg. 1),

(4) We provide identification, estimation, and sample influence results for all of the quantities relevant to the above framework (Props. 4, 5). We evaluate our approach on three real-world examples (Ex. 2-3) and provide new empirical insights into bias amplification.

Our work is related to the previous literature on causal fairness and the causal decompositions appearing in this literature [Zhang and Bareinboim, 2018b, Plečko and Bareinboim, 2024]. It is also related to previous literature on studying business necessity requirements through a causal lens [Kilbertus et al., 2017, Plecko and Bareinboim, 2024b]. However, our approach offers an entirely new causal decomposition into contributions from the true outcome $Y$ and the margin complement $M$. More broadly, our work is also related to the literature on fair decision-making, which analyzes how prediction scores impact the fairness of decisions [Chouldechova, 2017, Dwork et al., 2020, Chouldechova and Roth, 2018], or how disparities evolve over time [Liu et al., 2018]. Recent results also show that focusing purely on prediction, and ignoring decision-making aspects, may lead to inequitable outcomes and cause harm to marginalized groups [Plečko and Bareinboim, 2024, Nilforoshan et al., 2022, Plecko and Bareinboim, 2024a], highlighting a need to expand focus from narrow statistical definitions of fair predictions to a more comprehensive understanding of equity in algorithmic decisions. Still, many questions remain open in the context of fair decision-making, and more future works are required in this area. Finally, we mention that our work is also related to the literature on auditing and assessing fairness of decisions made by humans [Pierson et al., 2021, Kleinberg et al., 2018], and understanding how AI systems may help humans overcome their biases [Imai et al., 2023].

## 1.1 Preliminaries

We use the language of structural causal models (SCMs) [Pearl, 2000]. An SCM is a tuple $\mathcal{M} := \langle V, U, \mathcal{F}, P(u) \rangle$, where $V$, $U$ are sets of endogenous (observable) and exogenous (latent) variables, respectively, $\mathcal{F}$ is a set of functions $f_{V_i}$, one for each $V_i \in V$, where $V_i \leftarrow f_{V_i}(\mathrm{pa}(V_i), U_{V_i})$ for some $\mathrm{pa}(V_i) \subseteq V$ and $U_{V_i} \subseteq U$. The set $\mathrm{pa}(V_i)$ is called the parent set of $V_i$. $P(u)$ is a strictly positive probability measure over $U$. Each

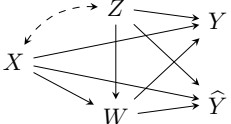

Figure 2: Standard Fairness Model.

SCM $\mathcal{M}$ is associated to a causal diagram $\mathcal{G}$ [Bareinboim et al., 2022] over the node set $V$ where $V_i \rightarrow V_j$ if $V_i$ is an argument of $f_{V_j}$, and $V_i \leftarrow\!\!\text{--}\!\!\rightarrow V_j$ if the corresponding $U_{V_i}, U_{V_j}$ are not independent. An instantiation of the exogenous variables $U = u$ is called a *unit*. By $Y_x(u)$ we denote the potential response of $Y$ when setting $X = x$ for the unit $u$, which is the solution for $Y(u)$ to the set of equations obtained by evaluating the unit $u$ in the submodel $\mathcal{M}_x$, in which all equations in $\mathcal{F}$ associated with $X$ are replaced by $X = x$. Throughout the paper, we assume a specific cluster causal diagram $\mathcal{G}_{\mathrm{SFM}}$ known as the standard fairness model (SFM) [Plečko and Bareinboim, 2024] over endogenous variables $\{X, Z, W, Y, \widehat{Y}\}$ shown in Fig. 2 (see also [Anand et al., 2023]). The SFM consists of the following: *protected attribute*, labeled $X$ (e.g., gender, race, religion), assumed to be binary; the set of *confounding* variables $Z$, which are not causally influenced by the attribute $X$ (e.g., demographic information, zip code); the set of *mediator* variables $W$ that are possibly causally influenced by the attribute (e.g., educational level or other job-related information); the *outcome* variable $Y$ (e.g., GPA, salary); the *predictor* of the outcome $\widehat{Y}$ (e.g., predicted GPA, predicted salary). The SFM also encodes the lack-of-confounding assumptions typically used in the causal inference literature. The availability of the SFM and the implied assumptions are a possible limitation of the paper, while we note that partial identification techniques for bounding effects can be used for relaxing them [Zhang et al., 2022].

## 2 Margin Complements

We begin by introducing a quantity that plays a key role in the results of this paper.

**Definition 1** (Margin Complement). *Let $U = u$ be a unit, and let $S$ denote a prediction score for a binary outcome $Y$. Let the subscript $C$ denote a counterfactual clause, so that $Z_C$ denotes a potential response. The margin complement $M$ of the score $S$ for the unit $U = u$ and threshold $t$ is defined as:*

$$M(u) = \mathbb{1}(S(u) \geq t) - S(u). \tag{4}$$

*A potential response of $M$, labeled $M_C$, is given by $M_C(u) = \mathbb{1}(S_C(u) \geq t) - S_C(u)$.*

In words, the margin complement for a unit $U = u$ represents the difference in the score after thresholding vs. the score that would happen naturally.

**Example 1** (Disparities in Hiring continued). *Consider the hiring example with the SCM in Eqs. 1-2 with $p_0 = 0.49, p_1 = 0.51$. The unit $(U_X, U_Y) = (0, 0)$ corresponds to a male applicant $(X(u) = 1)$ who was hired $(Y(u) = 1)$. We have $S(u) = p_1 = 0.51$, and $M(u) = \mathbb{1}(S(u) \geq 0.5) - S(u) = 1 - 0.51 = 0.49$. For this $u$, the margin complement indicates that the predicted outcome $\widehat{Y}(u) = \mathbb{1}(S(u) \geq 0.5)$ is 49% greater than the predicted probability $S(u)$. The same computation can be done for a female $X(u') = 0$, in which case $M(u') = \mathbb{1}(p_{x_0} \geq 0.5) - p_{x_0} = -0.49$, meaning that the predicted outcome $\widehat{Y}(u')$ is 49% smaller than the predicted probability $S(u')$.* $\square$

Given a prediction score $S$ and a threshold $t$, the margin complement $M$ tells us in which direction the thresholded version $\mathbb{1}(S(u) \geq t)$ moves compared to the score $S(u)$. A positive margin complement indicates that a thresholded predictor is larger than the probability prediction, and a negative margin complement the opposite. A similar reasoning holds for the potential responses of the margin complement $M_C$: we are interested in what the margin complement *would have been* for an individual $U = u$ under possibly different, counterfactual conditions described by $C$. As we demonstrate shortly, margin complements (and their potential responses) play a major role in explaining how inequities are generated between groups at the time of decision-making. In this section, our key aim is to analyze the optimal 0/1 predictor $\widehat{Y}$ and provide a decomposition of its total variation measure (TV, for short), defined as $\text{TV}_{x_0,x_1}(\widehat{y}) = P(\widehat{y} \mid x_1) - P(\widehat{y} \mid x_0)$. When working with the causal diagram in Fig. 2, we can notice that the TV measure comprises of three types of variations coming from $X$: the direct effect $X \to \widehat{Y}$, the mediated effect $X \to W \to \widehat{Y}$, and the confounded effect $X \leftarrow\!\!\dashrightarrow Z \to \widehat{Y}$. Our goal is to construct a decomposition of the TV measure that allows us to distinguish how much of each of the causal effects is due to a difference in the prediction score $S$, and how much due to margin complements $M$. To investigate this, we first introduce the known definitions of direct, indirect, and spurious effects from the causal fairness literature:

**Definition 2** ($x$-specific Causal Measures [Zhang and Bareinboim, 2018b, Plečko and Bareinboim, 2024]). *The $x$-specific {direct, indirect, spurious} effects of $X$ on $Y$ are defined as:*

$$x\text{-}DE_{x_0,x_1}(y \mid x) = P(y_{x_1,W_{x_0}} \mid x) - P(y_{x_0} \mid x) \tag{5}$$

$$x\text{-}IE_{x_1,x_0}(y \mid x) = P(y_{x_1,W_{x_0}} \mid x) - P(y_{x_1} \mid x) \tag{6}$$

$$x\text{-}SE_{x_1,x_0}(y) = P(y_{x_1} \mid x_0) - P(y_{x_1} \mid x_1). \tag{7}$$

Armed with these definitions, we can prove the following result:

**Theorem 1** (Causal Decomposition of Optimal 0/1 Predictor). *Let $\widehat{Y}$ be the optimal predictor with respect to the 0/1-loss based on covariates $X, Z, W$. Let $S$ denote the optimal predictor with respect to the $L_2$ loss. The total variation (TV, for short) measure of the predictor $\widehat{Y}$, written as $P(\hat{y} \mid x_1) - P(\hat{y} \mid x_0)$, can be decomposed into direct, indirect, and spurious effects of $X$ on the score $S$ and the margin complement $M$ as follows:*

$$TV_{x_0,x_1}(\widehat{y}) = x\text{-}DE_{x_0,x_1}(s \mid x_0) + x\text{-}DE_{x_0,x_1}(m \mid x_0) \tag{8}$$

$$- \left(x\text{-}IE_{x_1,x_0}(s \mid x_0) + x\text{-}IE_{x_1,x_0}(m \mid x_0)\right) \tag{9}$$

$$- \left(x\text{-}SE_{x_1,x_0}(s) + x\text{-}SE_{x_1,x_0}(m)\right). \tag{10}$$

The above theorem is the first key result of this paper. The disparity between groups with respect to the optimal 0/1-loss predictor, measured by $\text{TV}_{x_0,x_1}(\hat{y})$ can be decomposed into direct, indirect, and spurious contributions coming from (i) the optimal $L_2$-loss predictor $S$ (e.g., term $x\text{-DE}_{x_0,x_1}(s \mid x_0)$), and (ii) the margin complement $M$ (e.g., term $x\text{-DE}_{x_0,x_1}(m \mid x_0)$). This provides a unique capability since for each causal pathway (direct, indirect, spurious) the contribution coming from the probability prediction $S$ can be disentangled from the contribution coming from the optimization procedure itself (i.e., the rounding of the predictor). The former, as we will see shortly, is simply a representation of the bias already existing in the true outcome $Y$, whereas the latter represents a newly introduced type of bias that is the result of using an automated system. The contribution of the margin complement may act to both ameliorate or amplify an existing disparity, a point we investigate later on.

**Example 1** (Disparities in Hiring extended). *Consider the hiring example from Ex. 1 extended with a mediator $W$ indicating whether the applicant has a PhD degree ($W = 1$) or not ($W = 0$). Suppose*

*the augmented SCM is given by:*

$$X \leftarrow \mathbb{1}(U_X < 0.5) \tag{11}$$

$$W \leftarrow \mathbb{1}(U_W < 0.5 + \lambda X) \tag{12}$$

$$Y \leftarrow \mathbb{1}(U_Y < 0.1 + \alpha X + \beta W), \tag{13}$$

*and $P(U_X, U_W, U_Y)$ is such that $U_X, U_W, U_Y$ are independent Unif$[0,1]$ random variables. The coefficients satisfy the constraints $\alpha, \lambda > 0, 0.4 < \beta < \frac{0.9 - \alpha}{1 + \lambda}$. The optimal probability predictor is $S(x, w) = 0.1 + \alpha x + \beta w$. The TV measure of the optimal 0/1 predictor be computed as $TV_{x_0, x_1}(\hat{y}) = \frac{\alpha}{\beta} + \lambda$. Using Thm. 1 we can decompose it as:*

$$TV_{x_0, x_1}(\widehat{y}) = \underbrace{\alpha}_{x\text{-}DE_{x_0, x_1}(s|x_0)} + \underbrace{\frac{\alpha}{\beta} - \alpha}_{x\text{-}DE_{x_0, x_1}(m|x_0)} - \underbrace{(-\lambda\beta)}_{x\text{-}IE_{x_1, x_0}(s|x_0)} - \underbrace{(\lambda - \lambda\beta)}_{x\text{-}IE_{x_1, x_0}(m|x_0)} . \tag{14}$$

*Along the direct effect, there is a difference of $\alpha$ between the groups in the probability predictor $S(x, w)$ that is amplified by $\frac{\alpha}{\beta} - \alpha$ by the margin complement, and since $\beta < 1$ we always have bias amplification. Along the indirect effect, there is an initial difference of $\lambda\beta$, amplified by an additional $\lambda - \lambda\beta$, again amplifying bias. The decomposition provides a breakdown of the disparity of the optimal 0/1 predictor into its constitutive parts.* □

We next consider a crucial point mentioned above. The disparity observed in the optimal $L_2$ score $S$ is simply a representation of the disparity existing in the real world, under suitable causal assumptions:

**Theorem 2** (Relationship of $L_2$-score $S$ and Outcome $Y$ Decompositions). *Let $\mathcal{M}$ be an SCM compatible with the Standard Fairness Model, and let $S$ be the optimal $L_2$ prediction score. Then, the causal decompositions of the score $S$ and the true outcome $Y$ are symmetric, meaning that $x\text{-}IE_{x_1, x_0}(s \mid x_0) = x\text{-}IE_{x_1, x_0}(y \mid x_0)$, and $x\text{-}SE_{x_1, x_0}(s) = x\text{-}SE_{x_1, x_0}(y)$.*

$$x\text{-}DE_{x_0, x_1}(s \mid x_0) = x\text{-}DE_{x_0, x_1}(y \mid x_0) \tag{15}$$

$$x\text{-}IE_{x_1, x_0}(s \mid x_0) = x\text{-}IE_{x_1, x_0}(y \mid x_0) \tag{16}$$

$$x\text{-}SE_{x_1, x_0}(s) = x\text{-}SE_{x_1, x_0}(y). \tag{17}$$

Based on this result, we can see that the causal influences from $X$ on the true outcome $Y$ are equivalent to the causal influences from $X$ on the optimal prediction score $S$. Combining this insight with Thm. 1 leads to the following result:

**Corollary 3** (Causal Decomposition of Optimal 0/1 Predictor). *Under the Standard Fairness Model, the TV measure of the optimal 0/1-loss predictor $\widehat{Y}$ can be decomposed into contributions from the true outcome $Y$ and the margin complement $M$:*

$$TV_{x_0, x_1}(\widehat{y}) = x\text{-}DE_{x_0, x_1}(y \mid x_0) + x\text{-}DE_{x_0, x_1}(m \mid x_0) \tag{18}$$

$$- \big(x\text{-}IE_{x_1, x_0}(y \mid x_0) + x\text{-}IE_{x_1, x_0}(m \mid x_0)\big) \tag{19}$$

$$- \big(x\text{-}SE_{x_1, x_0}(y) + x\text{-}SE_{x_1, x_0}(m)\big). \tag{20}$$

This corollary provides an important insight compared to Thm. 1: the disparity in the optimal predictor $\widehat{Y}$ can be decomposed into the contribution inherited from the true outcome $Y$ and the contribution that arises from the optimization procedure (rounding of the predictor). In Appendix D, we describe two important extensions of the main results: (i) to the setting of suboptimal predictors; (ii) to the setting of predictors using group-specific thresholds, which is often the case with post-processing methods used in fair machine learning [Kamiran et al., 2012, Hardt et al., 2016].

## 3   Weak and Strong Business Necessity

In law, questions of fairness and discrimination can be interpreted based on the disparate treatment (DT) and disparate impact (DI) doctrines of Title VII of the Civil Rights Act of 1964 [Act, 1964]. The DT doctrine, when interpreted causally, can be seen as disallowing a direct type of effect from $X$ onto the outcome. The DI doctrine, however, is more broad, and may prohibit any from of discrimination (be it direct, indirect, or spurious) that results in a large disparity between groups. The core of this

---

**Algorithm 1:** Auditing Weak & Strong Business Necessity

---

**Input:** data $\mathcal{D}$, BN-Set $\subseteq \{\text{DE}, \text{IE}, \text{SE}\}$, BN-Strength, predictor $\widehat{Y}$, score $S$, SFM
**Output:** SUCCESS or FAIL of ensuring that disparate impact and treatment hold under BN

**1 foreach** $CE \in \{DE, IE, SE\}$ **do**
**2**     Compute the effects $x\text{-CE}(y)$, $x\text{-CE}(m)$, $x\text{-CE}(s)$, $x\text{-CE}(\widehat{y})$
**3**     **if** $CE \in$ *Strong-BN* **then**
**4**        Assert that $x\text{-CE}(s) = x\text{-CE}(\widehat{y})$, otherwise FAIL
**5**     **else if** $CE \in$ *Weak-BN* **then**
**6**        Assert that $x\text{-CE}(s) = x\text{-CE}(\widehat{y}) \wedge x\text{-CE}(m) = 0$, otherwise FAIL
**7**     **else**
**8**        Assert that $x\text{-CE}(\widehat{s}) = x\text{-CE}(\widehat{m}) = 0$, otherwise FAIL

**9 if** *not FAIL* **return** *SUCCESS*

---

doctrine is the notion of *business necessity* (BN), which allows certain variables correlated with the protected attribute to be used for prediction due to their relevance to the business itself [Els, 1993] (or more broadly the utility of the decision-maker). Based on the decomposition from Cor. 3, new BN considerations emerge:

**Definition 3** (Weak and Strong Business Necessity). *Let $\mathcal{M}$ be an SCM compatible with the Standard Fairness Model. Let CE denote a causal pathway (DE, IE, or SE), and let $x, x'$ be two distinct values of $X$. Let $x''$ be a third, arbitrary value of $X$. If a causal pathway does not fall under business necessity, then we require:*

$$x\text{-}CE_{x,x'}(s \mid x'') = x\text{-}CE_{x,x'}(m \mid x'') = 0. \tag{21}$$

*A pathway is said to satisfy weak business necessity if:*

$$x\text{-}CE_{x,x'}(s \mid x'') = x\text{-}CE_{x,x'}(y \mid x''), \ x\text{-}CE_{x,x'}(m \mid x'') = 0. \tag{22}$$

*A pathway is said to satisfy strong business necessity if:*

$$x\text{-}CE_{x,x'}(s \mid x'') = x\text{-}CE_{x,x'}(y \mid x''), \ \textit{while } x\text{-}CE_{x,x'}(m \mid x'') \textit{ takes any arbitrary value}. \tag{23}$$

The above definition distinguishes between three important cases, and sheds light on a new aspect of the concept of business necessity. According to the definition, there are three versions of BN considerations:

(1) A causal pathway is not in the BN set, and is considered discriminatory. In this case, both the contribution of the prediction score $S$ and the margin complement $M$ need to be equal to 0 (i.e., no discrimination is allowed along the pathway),

(2) A causal pathway satisfies weak BN, and is not considered discriminatory. In this case, the effect of $X$ on the prediction score $S$ needs to equal the effect of $X$ onto the true outcome $Y$ along the same pathway [Plecko and Bareinboim, 2024b]. However, the contribution of the margin complement $M$ along the pathway needs to equal 0.

(3) A causal pathway satisfies strong BN, and is not considered discriminatory. Similarly as for weak necessity, the effect of $X$ on $S$ needs to equal the effect of $X$ on $Y$, but in this case, the contribution of the margin complement $M$ is unconstrained.

The distinction between cases (2) and (3) opens the door for new regulatory requirements and specifications. In particular, whenever a causal effect is considered non-discriminatory, the attribute $X$ needs to affect $S$ to the extent to which it does in the real world. However, the system designer also needs to decide whether a difference existing in the predicted probabilities $S$ is allowed to be amplified (or ameliorated) by means of rounding. The latter point distinguishes between weak and strong BN, and should be a consideration of any system designer issuing binary decisions. In Alg. 1, we propose a formal approach for evaluating considerations of weak and strong BN for any input of a predictor $\widehat{Y}$ and a prediction score $S$.

# 4 Identification, Estimation, and Sample Influence

In Thm. 1 and Cor. 3 the observed disparity in the TV measure of the optimal 0/1 predictor is decomposed into its constitutive components. The quantities appearing in the decomposition are counterfactuals, and thus the question of *identification* of these quantities needs to be addressed. In other words, we need to understand whether these quantities can be uniquely computed based on the available data and the causal assumptions. The following is a positive answer:

**Proposition 4** (Identification and Estimation of Causal Measures). *Let $\mathcal{M}$ be an SCM compatible with the Standard Fairness Model, and let $P(V)$ be its observational distribution. The $x$-specific direct, indirect, and spurious effects of $X$ on the outcome $Y$, predictor $\widehat{Y}$, prediction score $S$, and the margin complement $M$ are identifiable (uniquely computable) from $P(V)$ and the SFM. Denote by $f(x, z, w)$ estimator of $\mathbb{E}[T \mid x, z, w]$, and by $\hat{P}(x \mid v')$ the estimator of the probability $P(x \mid v')$ for different choices of $v'$. For $T \in \{Y, \widehat{Y}, S, M\}$, the effects can be estimated as:*

$$x\text{-}DE^{\text{est}}_{x_0,x_1}(t \mid x_0) = \frac{1}{n} \sum_{i=1}^{n} [f(x_1, w_i, z_i) - f(x_0, w_i, z_i)] \frac{\hat{P}(x_0 \mid w_i, z_i)}{\hat{P}(x_0)} \tag{24}$$

$$x\text{-}IE^{\text{est}}_{x_1,x_0}(t \mid x_0) = \frac{1}{n} \sum_{i=1}^{n} f(x_1, w_i, z_i) \Big[ \frac{\hat{P}(x_0 \mid w_i, z_i)}{\hat{P}(x_0)} - \frac{\hat{P}(x_1 \mid w_i, z_i)}{\hat{P}(x_1 \mid z_i)} \frac{\hat{P}(x_0 \mid z_i)}{\hat{P}(x_0)} \Big] \tag{25}$$

$$x\text{-}SE^{\text{est}}_{x_1,x_0}(t) = \frac{1}{n} \sum_{i=1}^{n} f(x_1, w_i, z_i) \Big[ \frac{\hat{P}(x_1 \mid w_i, z_i)}{\hat{P}(x_1 \mid z_i)} \frac{\hat{P}(x_0 \mid z_i)}{\hat{P}(x_0)} - \frac{\hat{P}(x_1 \mid w_i, z_i)}{\hat{P}(x_1)} \Big]. \tag{26}$$

The proof of the proposition, together with the identification expressions for the different quantities can be found in Appendix B. We next define the sample influences for the different estimators:

**Definition 4** (Sample Influence). *The sample influence of the $i$-th sample on the estimator $x$-$CE^{\text{est}}$ of the causal effect CE is given by corresponding term in the summations in Eqs. 24-26. For instance, the $i$-th sample influence on $x$-$DE^{\text{est}}_{x_0,x_1}(t \mid x_0)$ is given by (and analogously for IE, SE terms):*

$$SI\text{-}DE(i) = [f(x_1, w_i, z_i) - f(x_0, w_i, z_i)] \frac{\hat{P}(x_0 \mid w_i, z_i)}{\hat{P}(x_0)}. \tag{27}$$

The sample influences tell us how each of the samples contributes to the overall estimator of the quantity. These sample-level contributions may be interesting to investigate from the point of view of the system designer, including identifying any subpopulations that are discriminated against. For direct sample influences, the following proposition can be proved:

**Proposition 5** (Direct Effect Sample Influence). *The SI-DE($i$) in Eq. 27 is an estimator of*

$$\mathbb{E}[T_{x_1, W_{x_0}} - T_{x_0} \mid x_0, z_i, w_i] \frac{P(w_i, z_i \mid x_0)}{P(w_i, z_i)}, \tag{28}$$

*where $\mathbb{E}[T_{x_1, W_{x_0}} - T_{x_0} \mid x_0, z_i, w_i]$ is the $(x_0, z_i, w_i)$-specific direct effect of $X$ on $T$.*

Prop. 5 demonstrates an important point – namely that the sample influences along the direct path are not just quantities of statistical interest, but also *causally* meaningful quantities. In particular, the influence of the $i$-th sample is proportional to the direct effect of the $x_0 \to x_1$ transition for the group of units $u$ compatible with the event $x_0, z_i, w_i$. The influence is further proportional to $P(w_i, z_i \mid x_0)/P(w_i, z_i)$ that measures how much more likely the covariates $z_i, w_i$ of the $i$-th sample are in the $X = x_0$ group (for which the discrimination is quantified) vs. the overall population. Therefore, practitioners also have a causal reason for investigating these sample influences.

# 5 Experiments

We analyze the MIMIC-IV (Ex. 2), COMPAS (Ex. 3), and Census (Ex. 4, Appendix C) datasets.

**Example 2** (Acute Care Triage on MIMIC-IV Dataset [Johnson et al., 2023]). *Clinicians in the Beth Israel Deaconess Medical Center in Boston, Massachusetts treat critically ill patients admitted to*

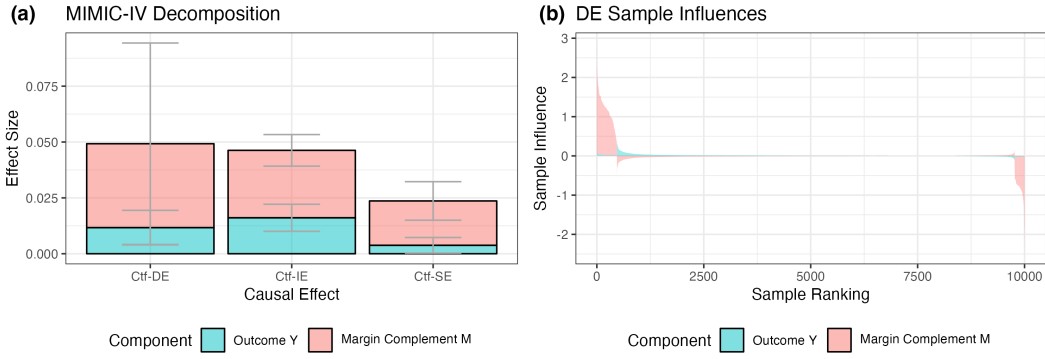

Figure 3: Causal decomposition from Cor. 3 and sample influence on the MIMIC-IV dataset.

*the intensive care unit (ICU). For all patients, various physiological and treatment information is collected 24 hours after admission, and the available data consists of (grouped into the Standard Fairness model): protected attribute $X$, in this case race ($x_0$ African-American, $x_1$ White), set of confounders $Z$ = {sex, age, chronic health status}, set of mediators $W$ ={lactate, SOFA score, admission diagnosis, $PaO_2/FiO_2$ ratio, aspartate aminotransferase}.*

*Clinicians are interested in patients who require closer monitoring. They want to determine the top half of the patients who are the most likely to (i) die during their hospital stay; (ii) have an ICU stay longer than 10 days. This combined outcome is labeled $Y$. These high-risk patients will remain in the most acute care unit. To predict the outcome, clinicians use the electronic health records (EHR) data of the hospital and construct score predictions $S$ and a binary predictor $\widehat{Y} = \mathbb{1}(S(x, z, w) > \text{Quant}(0.5; S))$ that selects the top half of the patients.*

*To investigate the fairness implications of the new AI-based system, they use the decomposition described in Cor. 3 to investigate different contributions to the resulting disparity. The decomposition is shown in Fig. 3(a), and uncovers a number of important effects. Firstly, along the direct effect, $x\text{-}DE_{x_0,x_1}(y)$ and $x\text{-}DE_{x_0,x_1}(m)$ are larger than 0, meaning that minority group individuals have a lower chance of receiving acute care (purely based on race). Along the indirect and spurious effects, the situation is different: $x\text{-}IE_{x_1,x_0}(y)$ and $x\text{-}SE_{x_1,x_0}(y)$ and their respective margin complement contributions are different from 0 and negative – implying that minority group individuals have a larger probability of being given acute care as a result of confounding and mediating variables. Finally, the direct effect sample influences (Fig. 3(b)) highlight that the margin complements are large for a small minority of individuals, requiring further subgroup investigation by the hospital team.* □

**Example 3** (Recidivism Prevention on the COMPAS Dataset [Larson et al., 2016])**.** *Courts in Broward County, Florida use machine learning algorithms, developed by a private company called Northpointe, to predict whether individuals released on parole are at high risk of re-offending within 2 years ($Y$). The algorithm is based on the demographic information $Z$ ($Z_1$ for gender, $Z_2$ for age), race $X$ ($x_0$ denoting White, $x_1$ Non-White), juvenile offense counts $J$, prior offense count $P$, and degree of charge $D$. The courts wish to know which individuals are highly likely to recidivate, such that their probability of recidivism is above 50%. The company constructs a prediction score $\hat{S}^{NP}$ and the court subsequently uses this for deciding whether to detain individuals at high risk of re-offending.*

*After a court hearing in which it was decided that the indirect and spurious effects fall under business necessity requirements, a team from ProPublica wishes to investigate the implications of using the automated predictions $\hat{S}^{NP}$. They obtain the relevant data and apply Alg. 1, with the results shown in Fig. 4. The team first compares the decompositions of the true outcome $Y$ and the predictor $\hat{S}^{NP}$ (Fig. 4(a)). For the spurious effect, they find that $x\text{-}SE_{x_1,x_0}(y)$ is not statistically different from $x\text{-}SE_{x_1,x_0}(\hat{s}^{NP})$, in line with BN requirements. For the indirect effects, they find that the indirect effect is lower for the predictor $\hat{S}^{NP}$ compared to the true outcome $Y$, indicating no concerning violations. However, for the direct effect, while the $x\text{-}DE_{x_0,x_1}(y)$ is not statistically different from 0, the predictor $\hat{S}^{NP}$ has a significant direct effect of $X$, i.e., $x\text{-}DE_{x_0,x_1}(\hat{s}^{NP}) \neq 0$. This indicates a violation of the fairness requirements determined by the court.*

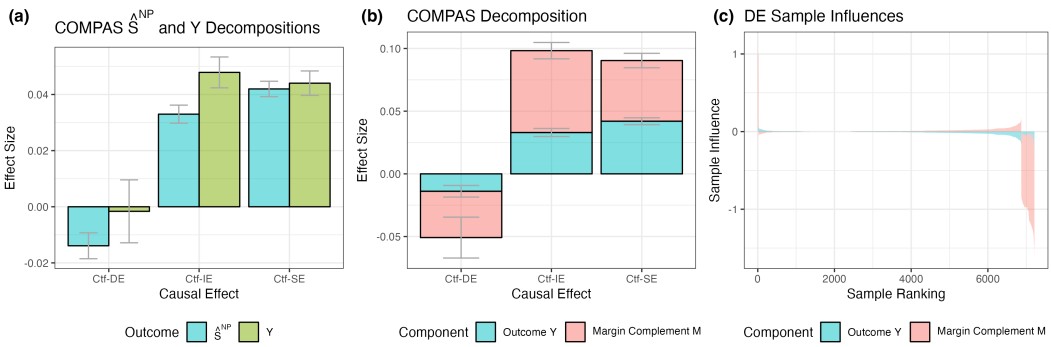

Figure 4: Application of Alg. 1 on the COMPAS dataset.

*After comparing the decompositions of $\hat{S}^{NP}$ and $Y$, the team moves onto understanding the contributions of the margin complements (Fig. 4b). For each effect, there is a pronounced impact of the margin complements. For the direct effect (not under BN), the non-zero margin complement contribution $x\text{-}DE_{x_1,x_0}(m) \neq 0$ represents a violation of fairness requirements. For the indirect and spurious effects, the ProPublica team realizes the court did not specify anything about margin complement contributions – based on this, for the next court hearing they are preparing an argument showing that the effects $x\text{-}IE_{x_1,x_0}(m)$ and $x\text{-}SE_{x_1,x_0}(m)$ are significantly different from $0$, thereby exacerbating the differences between groups. Finally, based on sample influences (Fig. 4c), they realize that the direct effect is driven by a small minority of individuals, and they decide to investigate this further.* $\square$

## 6   Conclusion

In this paper, we developed tools for understanding the fairness impacts of transforming a continuous prediction score $S$ into binary predictions $\widehat{Y}$ or binary decisions $D$. In Thm. 1 and Cor. 3 we showed that the TV measure of the optimal 0/1 predictor decomposes into direct, indirect, and spurious contributions that are inherited from the true outcome $Y$ in the real world, and also contributions from the margin complement $M$ (Def. 1) arising from the automated optimization procedure. This observation motivated new notions of *weak* and *strong* business necessity (BN) – in the former case, differences inherited from the true outcome $Y$ are allowed to be propagated into predictions or decisions, while any differences resulting from the optimization procedure are disallowed. In contrast, strong BN allows both of these differences and does not prohibit possible disparity amplification. In Alg. 1, we developed a formal procedure for assessing weak and strong BN. Finally, real-world examples demonstrated that the tools developed in this paper are of genuine importance in practice, since converting continuous predictions into binary decisions may often result in bias amplification in practice – highlighting the need for this type of analysis, and the importance of regulatory oversight.

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

# Technical Appendices for *Mind the Gap: A Causal Perspective on Bias Amplification in Prediction & Decision-Making*

The source code for reproducing all the experiments can be found in our Github code repository `https://github.com/dplecko/mind-the-gap`. The code is also included with the supplementary materials, in the folder `source-code`. All experiments were performed on a MacBook Pro, with the M3 Pro chip and 36 GB RAM on macOS 14.1 (Sonoma). All experiments can be run with less than 1 hour of compute on the above-described machine or equivalent.

## A  Proofs of Key Theorems

*Thm. 1 Proof:* Notice that by definition we have

$$M_C(u) = \mathbb{1}(S_C(u) \geq t) - S_C(u) \ \forall u \in \mathcal{U}. \tag{29}$$

Now, let $E$ be any observed event, and let $C$ be a counterfactual clause representing a possibly counterfactual intervention. Note that we can write:

$$\mathbb{E}[\widehat{Y}_C \mid E] \overset{\text{(def)}}{=} \sum_u \widehat{Y}_C(u) P(u \mid E) = \sum_u \mathbb{1}(S_C(u) \geq t) P(u \mid E) \tag{30}$$

$$= \sum_u \Big( \mathbb{1}(S_C(u) \geq t) - S_C(u) + S_C(u) \Big) P(u \mid E) \tag{31}$$

$$= \sum_u \Big( \mathbb{1}(S_C(u) \geq t) - S_C(u) \Big) P(u \mid E) + \sum_u S_C(u) P(u \mid E) \tag{32}$$

$$= \sum_u M_C(u) P(u \mid E) + \sum_u S_C(u) P(u \mid E) \tag{33}$$

$$= \mathbb{E}[M_C \mid E] + \mathbb{E}[S_C \mid E]. \tag{34}$$

Now, we can expand the TV measure of $\widehat{Y}$ as follows:

$$\text{TV}_{x_0,x_1}(\hat{y}) = \mathbb{E}[\widehat{Y} \mid x_1] - \mathbb{E}[\widehat{Y} \mid x_0] \tag{35}$$

$$= \mathbb{E}[\widehat{Y}_{x_1} \mid x_1] - \mathbb{E}[\widehat{Y}_{x_1} \mid x_0] \tag{36}$$

$$+ \mathbb{E}[\widehat{Y}_{x_1} \mid x_0] - \mathbb{E}[\widehat{Y}_{x_1, W_{x_0}} \mid x_0] \tag{37}$$

$$+ \mathbb{E}[\widehat{Y}_{x_1, W_{x_0}} \mid x_0] - \mathbb{E}[\widehat{Y}_{x_0} \mid x_0] \tag{38}$$

$$= \mathbb{E}[S_{x_1} \mid x_1] - \mathbb{E}[S_{x_1} \mid x_0] + \mathbb{E}[M_{x_1} \mid x_1] - \mathbb{E}[M_{x_1} \mid x_0] \tag{39}$$

$$+ \mathbb{E}[S_{x_1} \mid x_0] - \mathbb{E}[S_{x_1, W_{x_0}} \mid x_0] + \mathbb{E}[M_{x_1} \mid x_0] - \mathbb{E}[M_{x_1, W_{x_0}} \mid x_0] \tag{40}$$

$$+ \mathbb{E}[S_{x_1, W_{x_0}} \mid x_0] - \mathbb{E}[S_{x_0} \mid x_0] + \mathbb{E}[M_{x_1, W_{x_0}} \mid x_0] - \mathbb{E}[M_{x_0} \mid x_0] \tag{41}$$

$$= -x\text{-SE}_{x_1,x_0}(s) - x\text{-SE}_{x_1,x_0}(m) \tag{42}$$

$$- x\text{-IE}_{x_1,x_0}(s \mid x_0) - x\text{-IE}_{x_1,x_0}(m \mid x_0) \tag{43}$$

$$+ x\text{-DE}_{x_0,x_1}(s \mid x_0) + x\text{-DE}_{x_0,x_1}(m \mid x_0), \tag{44}$$

completing the proof. $\qquad\square$

*Thm. 2 Proof:* Let $\mathcal{M}$ be an SCM compatible with the SFM in Fig. 2 as per theorem assumption. Let the optimal $L_2$ prediction score $S$ be given by $S(x, z, w) = \mathbb{E}[Y \mid x, z, w]$. More precisely, we can let $f_S$ be the structural mechanism of $S$, taking $X, Z, W$ as inputs. The structural mechanism of $S$ is then given by:

$$f_S(x, z, w) = \mathbb{E}\left[Y \mid x, z, w\right]. \tag{45}$$

Therefore, the score $S$ is a deterministic function of $X, Z, W$, and we can add it to the standard fairness model as shown in Fig. 5a. Now, note that for a potential outcome $\mathbb{E}[S_{x, W_{x'}} \mid x'']$ we can

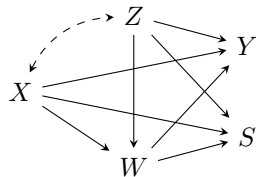
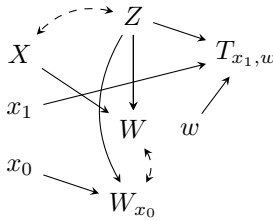

(a) SFM extended with predictor $S$.  (b) Counterfactual graph of the SFM.

Figure 5: Graphs used in proofs of Thm. 2 and Prop. 4.

write:

$$\mathbb{E}[S_{x,W_{x'}} \mid x''] = \sum_z \mathbb{E}[S_{x,W_{x'}} \mid x'', z] P(z \mid x'') \tag{46}$$

$$= \sum_{z,w} \mathbb{E}[S_{x,w,z} \mathbb{1}(W_{x'} = w) \mid x'', z] P(z \mid x'') \tag{47}$$

$$= \sum_{z,w} \mathbb{E}[S_{x,w,z} \mid x'', z] \mathbb{E}[\mathbb{1}(W_{x'} = w) \mid x'', z] P(z \mid x'') \tag{48}$$

$$= \sum_{z,w} \mathbb{E}[S_{x,w,z}] P(W_{x'} = w \mid x'', z) P(z \mid x'') \tag{49}$$

$$= \sum_{z,w} \mathbb{E}[Y_{x,w,z}] P(W_{x'} = w \mid x'', z) P(z \mid x'') \tag{50}$$

$$= \sum_{z,w} \mathbb{E}[Y_{x,w,z} \mathbb{1}(W_{x'} = w) \mid x'', z] P(z \mid x'') \tag{51}$$

$$= \sum_z \mathbb{E}[Y_{x,W_{x'}} \mid x'', z] P(z \mid x'') = \mathbb{E}[Y_{x,W_{x'}} \mid x''], \tag{52}$$

where Eq. 48 is using the independence $S_{x,z,w} \perp\!\!\!\perp W_{x'} \mid X, Z$, and Eq. 51 is using the independence $Y_{x,z,w} \perp\!\!\!\perp W_{x'} \mid X, Z$, both of which are implied by the standard fairness model extended with the node $S$ above. Based on the equality $\mathbb{E}[S_{x,W_{x'}} \mid x''] = \mathbb{E}[Y_{x,W_{x'}} \mid x'']$ for arbitrary values $x, x', x''$, the claim follows from the appropriate choices: for instance, taking $\{x = x_1, x' = x_0, x'' = x_0\}$ and $\{x = x_0, x' = x_0, x'' = x_0\}$, it follows that

$$x\text{-DE}_{x_0,x_1}(s) = \mathbb{E}[S_{x_1,W_{x_0}} \mid x_0] - \mathbb{E}[S_{x_0,W_{x_0}} \mid x_0] \tag{53}$$

$$= \mathbb{E}[Y_{x_1,W_{x_0}} \mid x_0] - \mathbb{E}[Y_{x_0,W_{x_0}} \mid x_0] = x\text{-DE}_{x_0,x_1}(y). \tag{54}$$

Similarly, analogous choices of $x, x', x''$ can be used to show the equality for indirect and spurious effects. $\qquad\square$

## B  Proofs related to Identification & Estimation

*Prop. 4 Proof:* Consider the $x$-specific {direct, indirect, spurious} effects of $X$ on a random variable $T$ given by:

$$x\text{-DE}_{x_0,x_1}(t \mid x_0) = P(t_{x_1,W_{x_0}} \mid x_0) - P(t_{x_0} \mid x_0) \tag{55}$$

$$x\text{-IE}_{x_1,x_0}(t \mid x_0) = P(t_{x_1,W_{x_0}} \mid x_0) - P(t_{x_1} \mid x_0) \tag{56}$$

$$x\text{-SE}_{x_1,x_0}(t) = P(t_{x_1} \mid x_0) - P(t_{x_1} \mid x_1). \tag{57}$$

We first demonstrate why each of the potential outcomes appearing is identifiable under the Standard Fairness Model. Consider the potential outcome $T_{x_1,W_{x_0}} \mid X = x_0$. For proving its identifiability, we make use of the counterfactual graph [Shpitser and Pearl, 2007] in Fig. 5b. We can expand

$P(T_{x_1, W_{x_0}} = t \mid x_0)$ as follows:

$$= \sum_{w} P(T_{x_1,w} = t, W_{x_0} = w \mid x_0) \quad \text{(Counterfactual Un-nesting)} \tag{58}$$

$$= \sum_{z,w} P(T_{x_1,w} = t, W_{x_0} = w \mid z, x_0) P(z \mid x_0) \quad \text{(Law of Total Probability)} \tag{59}$$

$$= \sum_{z,w} P(T_{x_1,w} = t \mid z, x_0) P(W_{x_0} = w \mid z, x_0) P(z \mid x_0) \quad (T_{x_1,w} \perp\!\!\!\perp W_{x_0} \mid Z, X) \tag{60}$$

$$= \sum_{z,w} P(T_{x_1,w} = t \mid z, x_0) P(W = w \mid z, x_0) P(z \mid x_0) \quad \text{(Consistency)} \tag{61}$$

$$= \sum_{z,w} P(T_{x_1,w} = t \mid z, w, x_0) P(W = w \mid z, x_0) P(z \mid x_0) \quad (T_{x_1,w} \perp\!\!\!\perp W \mid Z, X) \tag{62}$$

$$= \sum_{z,w} P(T_{x_1,w} = t \mid z, w, x_1) P(W = w \mid z, x_0) P(z \mid x_0) \quad (T_{x_1,w} \perp\!\!\!\perp X \mid Z) \tag{63}$$

$$= \sum_{z,w} P(T = t \mid z, w, x_1) P(W = w \mid z, x_0) P(z \mid x_0) \quad \text{(Consistency)} \tag{64}$$

$$= \sum_{z,w} P(t \mid z, w, x_1) P(w \mid z, x_0) P(z \mid x_0). \tag{65}$$

Using a similar but simplified argument, we also obtain that:

$$P(T_{x_1} = t \mid x_0) = \sum_{z} P(t \mid x_1, z) P(z \mid x_0), \tag{66}$$

and since $P(T_x = t \mid x) = P(t \mid x)$, all the terms in Eqs. 55-57 can be computed based on observational data.

We next move onto the estimation part of the proof. We again focus on the quantity $P(T_{x_1, W_{x_0}} = t \mid x_0)$. Suppose that we have:

$$f(x_1, z, w) \xrightarrow{P} \mathbb{E}[T \mid x_1, z, w] \ \forall z, w \tag{67}$$

$$\hat{P}(x_0 \mid w, z) \xrightarrow{P} P(x_0 \mid w, z) \ \forall z, w \tag{68}$$

$$\hat{P}(w, z) \xrightarrow{P} P(w, z) \ \forall w, z \tag{69}$$

$$\hat{P}(x_0) \xrightarrow{P} P(x_0) \tag{70}$$

where $f, \hat{P}(x_0 \mid w, z)$ are estimators, and $\hat{P}(w, z), \hat{P}(x_0)$ are the empirical distributions:

$$\hat{P}(w, z) = \frac{1}{n} \sum_{i=1}^{n} \mathbb{1}(Z_i = z, W_i = w) \tag{71}$$

$$\hat{P}(x_0) = \frac{1}{n} \sum_{i=1}^{n} \mathbb{1}(X_i = x_0). \tag{72}$$

Now, note that we can re-write the identification expression from Eq. 65 as:

$$\sum_{z,w} P(t \mid z, w, x_1) P(z, w) \frac{P(x_0 \mid z, w)}{P(x_0)}. \tag{73}$$

By a coupling of quantities in Eqs. 67-70 to a joint probability space and an application of the continuous mapping theorem we obtain that

$$\sum_{z,w} f(x_1, z, w) \hat{P}(z, w) \frac{\hat{P}(x_0 \mid z, w)}{\hat{P}(x_0)} \xrightarrow{P} \sum_{z,w} P(t \mid z, w, x_1) P(z, w) \frac{P(x_0 \mid z, w)}{P(x_0)}. \tag{74}$$

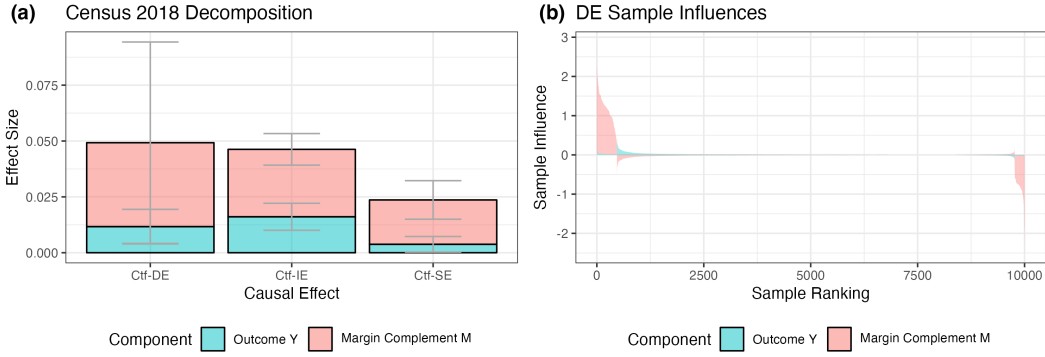

Figure 6: Causal decomposition from Cor. 3 and sample influence on the Census 2018 dataset.

Finally, note that we have

$$\sum_{z,w} f(x_1, z, w)\hat{P}(z, w)\frac{\hat{P}(x_0 \mid z, w)}{\hat{P}(x_0)} = \sum_{z,w} f(x_1, z, w)\Big(\sum_{i=1}^{n}\frac{\mathbb{1}(Z_i = z, W_i = w)}{n}\Big)\frac{\hat{P}(x_0 \mid z, w)}{\hat{P}(x_0)} \tag{75}$$

$$= \frac{1}{n}\sum_{i=1}^{n} f(x_1, z_i, w_i)\frac{\hat{P}(x_0 \mid z_i, w_i)}{\hat{P}(x_0)}. \tag{76}$$

The remaining identification expressions are derived based on the same technique, completing the proof of the proposition. □

*Prop. 5 Proof:* Consider the direct effect sample influence of the $i$-th sample given by

$$\text{SI-DE}(i) = [f(x_1, w_i, z_i) - f(x_0, w_i, z_i)]\frac{\hat{P}(x_0 \mid w_i, z_i)}{\hat{P}(x_0)}. \tag{77}$$

By assumption, we know that

$$f(x, z, w) \xrightarrow{P} \mathbb{E}[T \mid x_1, z, w] \; \forall x, z, w \tag{78}$$

$$\hat{P}(x_0 \mid w, z) \xrightarrow{P} P(x_0 \mid w, z) \; \forall z, w \tag{79}$$

$$\hat{P}(w, z) \xrightarrow{P} P(w, z) \; \forall w, z \tag{80}$$

$$\hat{P}(x_0) \xrightarrow{P} P(x_0) \tag{81}$$

Therefore, by a coupling of the above quantities to a joint probability space and an application of the continuous mapping theorem, we have that

$$\text{SI-DE}(i) \xrightarrow{P} \Big[\mathbb{E}[T \mid x_1, w_i, z_i] - \mathbb{E}[T \mid x_0, w_i, z_i]\Big] \times \frac{P(x_0 \mid w_i, z_i)}{P(x_0)}. \tag{82}$$

Note that

$$\frac{P(x_0 \mid w_i, z_i)}{P(x_0)} = \frac{P(x_0, w_i, z_i)}{P(w_i, z_i)P(x_0)} = \frac{P(w_i, z_i \mid x_0)}{P(w_i, z_i)}, \tag{83}$$

completing the proof of the proposition. □

## C   Census 2018 Example

In this appendix, we analyze the Census 2018 dataset, demonstrating an application in the context of labor and salary decisions:

**Example 4** (Salary Increase on the Census 2018 Dataset [Plečko and Bareinboim, 2024]). *The United States Census of 2018 collected broad information about the US Government employees, including demographic information $Z$ ($Z_1$ for age, $Z_2$ for race, $Z_3$ for nationality), gender $X$ ($x_0$ female, $x_1$ male), marital and family status $M$, education information $L$, and work-related information $R$. The US Government wishes to use this data prospectively to decide whether a new employee should receive a bonus starting package upon signing the employment contract. To determine which employees should receive such a bonus, a Government department decides to predict which of the employees should earn a salary above the median (which is $50,000/year). They construct a machine learning prediction score $S$ that predicts above-median earnings ($Y$), and the department decides to allocate the bonus to all employees who are predicted to be above-median earners with a probability greater than 50% (i.e., $\widehat{Y} = \mathbb{1}(S \geq 0.5)$).*

*A team of investigators within the Government is in charge of assessing what the impacts of this AI system are on the gender pay gap. They collect the required data and perform the decomposition from Cor. 3, shown in Fig. 6(a). The decomposition indicates strong direct effects $x\text{-}DE_{x_0,x_1}(y)$ and $x\text{-}DE_{x_0,x_1}(m)$, which imply that men are more likely to receive a starting bonus than women. The indirect effects $x\text{-}IE_{x_1,x_0}(y)$, $x\text{-}IE_{x_1,x_0}(m)$ are both negative, with the latter effect of the margin complement not being significant. These effects, however, also mean that men are more likely to receive a bonus due to mediating variables. Finally, for the spurious effects, the effects are not significantly different from $0$. Based on these findings, the team decided to return the predictions to the original department, with the requirement that the direct effect of gender on the margin complement, $x\text{-}DE_{x_0,x_1}(m)$, must be reduced to $0$, to avoid any possibility of bias amplification.* $\square$

## D   Extending Thm. 1 & Cor. 3

In this appendix, we discuss two important extensions of the results presented in the main paper. First, we note that Cor. 3 provides a decomposition of the TV measure for the optimal 0/1 predictor. In general, data analysts may be interested in analyzing suboptimal predictors using the same set of tools, which is discussed first (Appendix D.1). Second, we note that the results of the paper consider thresholded predictors $\widehat{Y} = \mathbb{1}(S \geq t)$. In practice, it may be desirable to analyze predictors that use *group-specific thresholds*, e.g., those of the from

$$\widehat{Y} = \mathbb{1}(S \geq t_x), \tag{84}$$

where $t_x$ is a group-specific threshold that may differ for $X = x_0$ and $X = x_1$ groups. This is the second extension we consider, and it is discussed in Appendix D.2.

### D.1   Suboptimal Predictors

We now consider how Cor. 3 could be applied to analyze suboptimal predictors. Cor. 3 was based on Thm. 2, which showed that the causal effects of $X$ on $Y$ were equal to the causal effects of $X$ on the optimal prediction score $S$. In practice, if a suboptimal prediction score $\tilde{S}$ is used, the symmetry shown in Thm. 2 can no longer be used. Therefore, if a predictor $\tilde{Y}$ is based on a suboptimal score $\tilde{S}$, the result of Cor. 3 cannot be applied directly.

Still, in this case, a variation of Cor. 3 can be used, stated in the following theorem:

**Theorem 6** (Decomposing Suboptimal Predictors). *Let $\tilde{Y}$ be any thresholded predictor based on a prediction score $\tilde{S}$, and let $\tilde{M}$ be its margin complement. Under the Standard Fairness Model in Fig. 2, the TV measure of the predictor $\tilde{Y}$ can be decomposed into contributions from the true outcome $Y$, the suboptimality of $\tilde{S}$, and the margin complement $\tilde{M}$:*

$$TV_{x_0,x_1}(\tilde{y}) = x\text{-}DE_{x_0,x_1}(y \mid x_0) + (x\text{-}DE_{x_0,x_1}(\tilde{s} \mid x_0) - x\text{-}DE_{x_0,x_1}(s \mid x_0)) + x\text{-}DE_{x_0,x_1}(\tilde{m} \mid x_0) \tag{85}$$

$$- \left( x\text{-}IE_{x_1,x_0}(y \mid x_0) + (x\text{-}IE_{x_1,x_0}(\tilde{s} \mid x_0) - x\text{-}IE_{x_1,x_0}(s \mid x_0)) + x\text{-}IE_{x_1,x_0}(\tilde{m} \mid x_0) \right) \tag{86}$$

$$- \left( x\text{-}SE_{x_1,x_0}(y) + (x\text{-}SE_{x_1,x_0}(s) - x\text{-}SE_{x_1,x_0}(\tilde{s})) + x\text{-}SE_{x_1,x_0}(m) \right). \tag{87}$$

Recall that Cor. 3 was a two-way decomposition along each causal pathway, into: (i) the contribution arising from the true outcome $Y$; (ii) contribution from the thresholding, i.e., the margin complement $M$. In Thm. 6, the decomposition is a three-way one. First, as before, there is a contribution of the true outcome $Y$ along the causal pathway in question. Second, there is a contribution of suboptimality

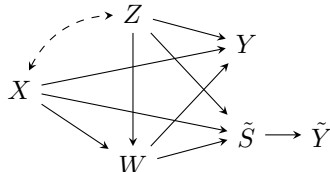
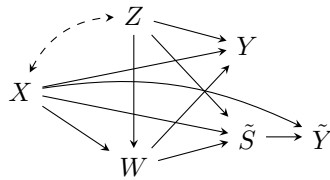

(a) SFM extended with predictor $S$.   (b) Counterfactual graph of the SFM.

Figure 7: Graphs used to understand predictors with group-specific thresholds.

of $\tilde{S}$ compared to $S$, which is captured by comparing the effects of $X$ on $\tilde{S}$ and $S$ along the causal pathway. Finally, there is also the contribution of thresholding, along the margin complement $\tilde{M}$ of $\tilde{S}$. Therefore, when considering a suboptimal predictor, three is an additional term in the decomposition (for each causal pathway) that measures the suboptimality of the prediction score $\tilde{S}$.

### D.2 Predictors with Group-specific Thresholds

In the main text and Appendix D.1, we considered classifiers of the form

$$\tilde{Y} = \mathbb{1}(\tilde{S} \geq t), \tag{88}$$

where $\tilde{S}$ is an arbitrary prediction score. In practice, when fairness considerations are taken into account, the decision-maker may use a group specific threshold, i.e., a classifier of the form:

$$\tilde{Y} = \begin{cases} \mathbb{1}(\tilde{S} \geq t_{x_0}) \text{ if } X = x_0 \\ \mathbb{1}(\tilde{S} \geq t_{x_1}) \text{ if } X = x_1. \end{cases} \tag{89}$$

In Eq. 89, different thresholds $t_{x_0}, t_{x_1}$ are used for groups $x_0, x_1$, respectively. The usage of group-specific thresholds is also widespread in the fair ML literature, for instance Kamiran et al. [2012] uses a post-processing approach with group-specific thresholds to construct a predictor that satisfies demographic parity. Similarly, [Hardt et al., 2016] uses a post-processing approach with group-specific thresholds for achieving equality of odds. In Fig. 7a, we present the causal diagram that corresponds to a single threshold setting, as in Eq. 88. We note in this case that all of the effects from $X$ to $\tilde{Y}$ (direct, indirect, spurious), are mediated by the prediction score $\tilde{S}$. When considering group-specific thresholds, represented graphically in Fig. 7b, clearly the predictor $\tilde{Y}$ needs to also take $X$ as an input, on top of $\tilde{S}$. Therefore, allowing for group-specific thresholds results in an additional direct effect $X \to \tilde{Y}$. Motivated by this definition, we introduce the following notion:

**Definition 5** (Group-specific Threshold Direct Effect). *Let $\tilde{Y}$ be a thresholded predictor with group-specific thresholds, based on a prediction score $\tilde{S}$. The group-specific threshold direct effect for a unit $U = u$ is defined as:*

$$u\text{-}DE_{x_0,x_1}^{\text{GST}}(\tilde{y}) = \tilde{Y}_{x_1, W_{x_0}}(u) - \tilde{Y}_{x_0, \tilde{S}_{x_1}, W_{x_0}}(u) \tag{90}$$

$$= \mathbb{1}(\tilde{S}_{x_1, W_{x_0}}(u) \geq t_{x_1}) - \mathbb{1}(\tilde{S}_{x_1, W_{x_0}}(u) \geq t_{x_0}). \tag{91}$$

*The $x$-specific group-specific direct effect is defined as:*

$$x\text{-}DE_{x_0,x_1}^{\text{GST}}(\tilde{y} \mid x) = \mathbb{E}[u\text{-}DE_{x_0,x_1}^{\text{GST}}(\tilde{y}) \mid X = x]. \tag{92}$$

The unit-level group-specific threshold direct effect captures the effect of a $X = x_0 \to X = x_1$ change along the pathway $X \to \tilde{Y}$. It measures if a specific unit $U = u$ is classified differently when considering different thresholds $t_{x_1}, t_{x_0}$. The $x$-specific version of the effect simply averages the unit-level effect across all units compatible with $X = x$. Armed with the above definition, in the following result, we provide a new decomposition that can explicitly handle analysis of predictors with group-specific thresholds:

**Theorem 7** (Causal Decomposition of Predictor with Group-specific Threshold). *Let $\tilde{Y}$ be a thresholded predictor with group-specific thresholds as in Eq. 89, based on a prediction score $\tilde{S}$. Let $\tilde{M}$ be*

*the margin complement of $\tilde{S}$ with respect to the threshold $t_{x_0}$ of the $X = x_0$ group. The TV measure of $\tilde{Y}$ can be decomposed as:*

$$TV_{x_0,x_1}(\tilde{y}) = x\text{-}DE_{x_0,x_1}^{\text{GST}}(\tilde{y} \mid x_0) + x\text{-}DE_{x_0,x_1}(\tilde{s} \mid x_0) + x\text{-}DE_{x_0,x_1}(\tilde{m} \mid x_0) \tag{93}$$

$$- \left(x\text{-}IE_{x_1,x_0}(\tilde{s} \mid x_0) + x\text{-}IE_{x_1,x_0}(\tilde{m} \mid x_0)\right) \tag{94}$$

$$- \left(x\text{-}SE_{x_1,x_0}(\tilde{s}) + x\text{-}SE_{x_1,x_0}(\tilde{m})\right). \tag{95}$$

As one can see, the decomposition in Thm. 7 is similar to that appearing in Thm. 1. However, there is an additional term $x\text{-}DE_{x_0,x_1}^{\text{GST}}(\tilde{y} \mid x_0)$ appearing, which is the consequence of considering group-specific thresholds. The above theorem provides the first result establishing that, causally, post-processing methods with group-specific thresholds change only the direct effect of $X$ on $\tilde{Y}$.

