# OpenReview forum: "Mind the Gap: A Causal Perspective on Bias Amplification in Prediction & Decision-Making"
_NeurIPS.cc/2024/Conference — NeurIPS 2024 poster_

### Official Review · Reviewer_ZV8n · 2024-07-12

**Soundness:** 2
**Presentation:** 1
**Contribution:** 2
**Rating:** 6
**Confidence:** 2

**Summary:**

This paper studies the impact of downstream thresholding operations on continuous (and possibly fair) prediction scores. The paper argues that inappropriate thresholding can amplify or ameliorate the disparity in predictive performance across groups defined by the protected attribute. Using a causal framework, the paper provides a methodology to separate the disparities in the original predictive score from the disparities introduced due to thresholding.

**Strengths:**

1. One of the motivations of the paper is to understand the downstream utility of fair predictions. In that regard, I appreciate the more practical focus on investigating the barriers faced by popular fairness during real-world implementations.
2. The disentanglement of different kinds of business necessity requirements seems interesting by itself and allows for potentially meaningful connections between legal and empirical notions of fairness (although I do have some questions on it that are noted later).

**Weaknesses:**

1. The writing and presentation are often hard to follow and various parts of the analysis seem under-explained, which makes evaluating the paper’s claims difficult. For example, Definition 1 defines $M$ using a generic $t$-value, whereas the proof of Theorem 1 in the appendix, if I understand correctly, directly uses $t=1/2$. If the theorem is based on a specific $t$ value, then that should be made clear in the theorem and the surrounding text in the main body.\
  Similarly, when discussing “strong business necessity”, there’s very little explanation of what it means for a causal pathway to be “unconstrained”. With a lack of a clear explanation of this concept, I am not totally sure about the difference between weak and strong BN.

2. Related to thresholding, the paper never goes into the details of how a certain threshold is chosen. Again, seems like Theorem 1’s proof and Example 1 use $t=1/2$, but that is not the only possible choice of threshold that can be used in practice. In fact, if all the claims in the paper are based on the explicit assumption that $t=1/2$, then I don’t see the claims being generalizable to other threshold classifiers and also not applicable to various kinds of classifiers used in practice.

3. Additionally, there is a wide post-processing fairness literature that essentially proposes methods to choose group-specific thresholds so that the biases from training data and prediction scores do not propagate to the final binary predictions (e.g., Hardt et al. 2016 and related papers). I would strongly recommend an expanded discussion on this related work, especially comparing the proposed approach to other prior fairness works on choosing appropriate thresholds to guarantee outcome fairness.

4. Minor points and typos.\
    a. Potential missing word in Line 73 around “…ramifications more”.\
    b. In Example 1, seems like the outcome should be 0 and 1, and not $y_0$ and $y_1$.\
    c. Line 138, function $pa$ is not defined. I am assuming it means “parent variable” but needs to be properly introduced and described.

**Questions:**

The main questions I have are related to the above points.
1. Does the paper primarily focus on $t=1/2$ or are the results applicable for other threshold values as well?
2. Do the issues associated with margin complement persist even when after using post-processing approaches to achieve outcome fairness (like those introduced in Hardt et al. 2016)?
3. What’s meant by a causal pathway being “unconstrained” when defining strong BN?

**Limitations:**

Some limitations are acknowledged but the paper could do a better job of expanding on the limitations related to the full knowledge of structural causal models.

---

> ### Author Rebuttal · Authors · 2024-08-07
>
> Authors: We thank the reviewer for the time spent on our submission. We would like to state clearly that the results of the paper are not at all constrained to the $t = 1/2$ setting and work for any value of $t$. We therefore hope the reviewer can reconsider the contributions in light of this.
>
> (W1: threshold $t$ value) Thank you for giving us the opportunity to clarify this point. The proof of Theorem 1 was stated for $t=1/2$, but $t=1/2$ can be replaced with an arbitrary threshold $t$, which is now done in the manuscript. In other words, the method can be used for classifiers with arbitrary thresholds. Here, the reviewer is correctly noting that the notion of margin complement changes. We are no longer computing the margin complement with respect to 1/2, but with respect to an arbitrary threshold t. However, the proof remains the same for the arbitrary threshold, and the interpretation of the decomposition remains very similar.
>
> (W2: Limited generality) Considering (W1) above, we note that the concern raised by the reviewer is not a limitation of the proposed method and the method can be used in general. Furthermore, we also note that our goal is not to suggest how to choose a specific threshold, but to provide a diagnostic tool for an arbitrary choice of the threshold.
> Please also note that in the experimental section, in the MIMIC-IV example, we are using a threshold different from 1/2. In particular, we are using a threshold that is the median of the predicted probabilities, which is about 0.15 in the data. Still, the analysis yields important insights even in this case, highlighting the generality of the method in practice.
>
> We hope that this addresses the key concern of the reviewer.
>
> (W3: post-processing methods) Thanks for asking this, this is a really good question. First, we wish to note that (Hardt et. al., 2016) are attempting to provide a post-processing method to satisfy equality of odds (i.e., constructing fair predictions). In our work, we are trying to offer a diagnostic tool, that quantifies the impact of thresholding on the causal fairness measures. In other words, we are solving a slightly different problem (which would be called bias detection/quantification).
>
> Having said that, the reviewer raises an excellent point. Our analysis is based on classifiers with a single threshold, whereas (Hardt et. al., 2016) and many other methods use two thresholds, one for each group.  In the paper, we now added an appendix that handles the following question: is it possible to adapt the diagnostic tool in this paper to a thresholded classifier in which the threshold $t$ depends on $X$, i.e., $t_x$?
>
> The answer to this is yes! Please refer to point (P2) of the main response for an updated decomposition of the TV measure in case of group-specific thresholds. In this new decomposition, we see that group-specific thresholds change only the direct effect of $X$ on the predictor while the indirect and spurious effects remain invariant. There is a new explicit term that quantifies the direct effect of using a group-specific threshold (see (P2) of main response for the exact expression). We also remark that this gives the first result establishing that, causally, post-processing methods with group-specific thresholds change only the direct effect of $X$ on the predictor, and gives an explicit expression for this direct effect. We think this makes the paper substantially stronger, and thank the reviewer for pointing us in the right direction! Our method can now handle the post-processing approach in (Hardt et. al, 2016) and many other similar methods in the literature.
>
> (W4: typos) Thanks a lot for catching this. We have now fixed all of them!
>
> (Q1: t=1/2 or not?) The paper can handle an arbitrary threshold $t$, please see responses (W1), (W2).
>
> (Q2: post-processing relationship) Please see (W3) for an answer.
>
> (Q3: Unconstrained in Strong BN) Thanks for asking this, we have now clarified it in the text accordingly. By unconstrained we mean that the causal effect $x\\text{-CE}(m)$ can take an arbitrary value $c$, instead of being constrained to $0$ like in the weak BN setting. We hope this clarification in the writing helps address the question. Please let us know!

---

> > ### Comment · Reviewer_ZV8n · 2024-08-11
> >
> > Thanks for the detailed response and clarifications! I am satisfied with the response on the ability to handle arbitrary thresholds and will increase my score.
> > I also appreciate the response on group-specific thresholds, but agree with Reviewer bCLY that the discussion on prior related work needs to be expanded and more thorough.

---

> > > ### Author Response · Authors · 2024-08-11
> > > **Thank you for constructive feedback**
> > >
> > > We wish to thank the reviewer for the opportunity to engage and provide important clarifications on the manuscript. We also appreciate you sharing constructive suggestions, which we took very seriously, and which helped to improve our manuscript and led us to a new result.
> > >
> > > Related to the discussion on previous literature, as we mentioned to Reviewer bCLY, we already revised the discussion of related literature in fair decision-making. We added the suggested references within this context, and expanded the discussion on related works accordingly.
> > >
> > > We once again thank the reviewer for the constructive review process.

---

### Official Review · Reviewer_bCLY · 2024-07-14

**Soundness:** 2
**Presentation:** 2
**Contribution:** 3
**Rating:** 6
**Confidence:** 3

**Summary:**

The paper studies how much thresholding a predictor affects the disparity in the decisions according to sensitive attributes and formalizes new notions of business necessity based on the causal graph of features, outcome and prediction function.

**Strengths:**

- Understanding the amplification of bias along the machine learning and decision making pipeline is an important topic.

- There are some very interesting ideas that could have benefited from better execution in the paper. Having some technical definition of business necessity makes claims of BN testable in the real world, and could have policy impact.

- A relatively comprehensive attempt to study the topic. Results include decomposition, identification, with proofs, as well as examples/experiments. With some significant improvements, it will be a nice contribution.

**Weaknesses:**

- Technical writing needs improvement and some key definitions are clearly wrong.
(1) Looking at Definition 3, weak BN implies strong BN according to this definition. Why is the condition for weak BN stronger than for strong BN? I was also unable to make sense of the rest of this definition, e.g. part 1.
(2) some quantities/notation are not properly defined. E.g. in definition 3, what are s, x'', m, x', y, and where are they defined? There should be no reasonable doubt about what these letters refer to. Same comment holds for Theorem 1 (what is m, s?), Theorem 2 and corollary 2.
(3) Please use the notation :=, or similar, to denote definitional equivalence (rather than an equality _claim_).

- Theorem 1 and 2 appear to be basic, as far as I can tell from the definitions and the proof. I'm not sure why the decomposition, which follows from definition, needs to be presented as a main theorem. The main novelty appears to be the definition. If this is the case, you might consider making this clear to the reader instead of presenting something as more complex that it needs to be. If this is not the case, is there a way you can make the technical contribution clearer in the proof and writing?

- Example 1 is not a reasonable example. In this case (where you ONLY have gender on which to make a definition, and there is practically no signal from it), there would be no reason to use a predictor at all.

-Related work: The discussion of closely related work is sparse. The connection with the causal fairness literature is relegated to a few sentences in the introduction, and there is no meaningful discussion of the contributions of prior work. Please elaborate on the connection between your definitions of business need and causal fairness?

Lack of citations and discussion of prior work. The paper states "Most of the literature in fair machine learning
4 focuses on defining and achieving fairness criteria in the context of prediction,
5 while not explicitly focusing on how these predictions may be used later on in
6 the pipeline."

This is an inaccurate representation of the literature. Even one of the cited works Chouldechoulva (2016) has an entire section on how scores impact decisions. Minimizing prior work on fairness in decision making is at best unhelpful to the reader, and at worst, misinforms.

For prior work on fairness in decision making (indeed, how predictions are used later in the pipeline), also see the following and the cited works within:

Liu, L. T., Dean, S., Rolf, E., Simchowitz, M., & Hardt, M. (2018, July). Delayed impact of fair machine learning. In International Conference on Machine Learning (pp. 3150-3158). PMLR.

Chouldechova, Alexandra, and Aaron Roth. "The frontiers of fairness in machine learning." arXiv preprint arXiv:1810.08810 (2018).

Dwork, Cynthia, Christina Ilvento, and Meena Jagadeesan. "Individual fairness in pipelines." arXiv preprint arXiv:2004.05167 (2020).

**Questions:**

1. What is the key technical contribution in Theorem 1 and 2, other than the novelty of the definitions (which allows you to write TV in a new, "human interpretable" way)?
2. What is the connection between your definitions of business need and counterfactual fairness (e.g. Kusner at al 2017)?
3. Please explain the corrected version of Definition 3 and what is the intended use case for each of the three definitions?
4. What is the potential negative societal impact of using the weak and strong business need definitions in a regulatory context?

**Limitations:**

Incompletely addressed impact of weak and strong business need definitions.

---

> ### Author Rebuttal · Authors · 2024-08-07
>
> Authors: We thank the reviewer for the detailed review. We would like to draw the attention of the reviewer to some misunderstandings. The tone of the review seems much harsher than what we are used to in a venue such as NeurIPS and also how we perform our own reviews, always very respectfully. For instance, calling a definition “clearly wrong”, whereas the definition is correct, calling a theorem basic, saying that an example is not reasonable, comes across as harsh and discouraging. Even if there is a suspected mistake, it may be better to point it out politely, since the authors (including us) are always trying their best when writing. Thus, we hope that the reviewer can reconsider the assessment of the paper based on our detailed response below, especially because the reviewer seems to appreciate some genuine novelty in the paper, and we believe the response below clarifies the existing misunderstandings.
>
> (W1: Technical writing) Definition 3 is correct and “Weak business necessity” is indeed a special case of “Strong business necessity”. The concept of BN allows the predictor designer more flexibility in designing the predictor (fewer constraints). Therefore, strong BN (or stronger/higher flexibility) is indeed a more broad concept than weak BN (or weak/lower flexibility). More concretely, a predictor designer given the freedom to choose any arbitrary value for the $x-CE(m)$ effect, could choose to set this value to $0$ (and thus satisfy the weak BN requirement). However, under strong BN, they do not have to do so necessarily. We hope this clarifies a core confusion.
>
> Furthermore, we would like to clarify some of the notation. Indeed, in Definition 3, we now explicitly state that $x, x’$ are two distinct values of $X$, whereas $x^{’’}$ is an arbitrary value of $X$. We thank the reviewer for pointing this shortcoming out, which is now corrected.
> Regarding $y, m, s$ notation: here, we are following the common notation in the graphical approach to causality. The small letters indicate the random variable which is integrated out, in line with the previous works on causal fairness and the causality textbook (Pearl, 2000).
>
> (W2: Theorems are “basic”) We do not see the mentioned theorems as basic, and view this as a harsh assessment. While the definitions are important, it is apriori not clear that these definitions interact in such a nice & non-parametric way with the decomposition of the TV measure (Theorem 1). Furthermore, Theorem 2 requires replacing the true outcome $Y$ with $S$ in the causal diagram. This, even conceptually, is a non-trivial step that has not been done before, and requires specific causal assumptions (such as the SFM), and may not always hold. We invite the reviewer to check the counterfactual graph in Figure 5(b), and then re-assess whether the claim is trivial. Thus, we believe the results are non-trivial. Furthermore, we also do not think we made any attempt to present the results as more complex than they are. The statements and proofs are stated as clearly and concisely as possible.
>
> To draw a parallel with some well-known results, think about the Pearl’s decomposition of the total effect (TE) into natural direct and indirect effects (NDE, NIE). While this decomposition is a consequence of the definitions of NDE, NIE, it is by no means trivial, and required almost a decade to appear after the first notions of direct and indirect effects were considered.
>
> (W3: Example 1 is not reasonable) Example 1 is used to illustrate the core concept of the paper in the simplest possible, two-variable case. We now explicitly state this in the introduction.
> A data scientist who has access to just these two variables (and perhaps does not even know the meaning of $X$), may simply implement a predictor without checking the predictive value, right? Once again, we are not trying to argue that the example reflects real-world practice (now stated explicitly before the example), but rather illustrate the concepts in the simplest form. Please see the Experiments section for more realistic, practical examples.
>
> (W4: Related work) Thanks for the suggested references. We have now expanded the part on related work, and reference the works you have shared, referring to “notable exceptions investigating the decision-making aspects of fairness”.
>
> However, we do believe it is true that most of the literature looks at prediction, which is also noted in the  second reference that you shared. To be clear, our goal is certainly not to diminish any of the contributions in this area, but rather to argue that more works are needed in this direction. We hope that the addition of the suggested references, and the better placement of the work in a broader context helps address this concern. Please let us know.
>
> ---
>
> (Q1: Key technical contribution) Please see (W2) for a detailed response.
>
> (Q2: Connection with counterfactual fairness) This is a great question! It has been shown in the causal fairness literature that counterfactual fairness is simply a notion that constrains the _total (causal) effect)_ of $X$ on $Y$. When doing so, counterfactual fairness does not distinguish between, for instance, direct and indirect effects. As a consequence, counterfactual fairness inherently does not have the flexibility to model business necessity requirements, since it considers a single pathway (total causal effect). We hope this answers the question.
>
> (Q3: Definition 3) Please check the response (W1), the definition is correct. Regarding the intended use cases, please see details in global review response (P3).
>
> (Q4: Societal impact) This is another great question. In terms of negative consequences, one may argue that imposing weak BN or no BN constraints may harm the utility of the classifier, which may cause harm in specific applications. However, this represents a trade-off between utility and fairness, which is acknowledged in the literature. We now added an explicit discussion on this in the paper.

---

> ### Author Response · Authors · 2024-08-12
>
> Dear Reviewer bCLY,
>
> In the spirit of the constructive discussion period we usually have at NeurIPS, we would like to double-check if there are any issues that were not sufficiently addressed in our rebuttal. We would be happy to engage and elaborate further on any questions that you may have.
>
> Thank you again for the time spent reading our work,
>
> Authors of paper #3972

---

### Official Review · Reviewer_1zLn · 2024-07-15

**Soundness:** 3
**Presentation:** 2
**Contribution:** 3
**Rating:** 7
**Confidence:** 2

**Summary:**

The paper investigates how thresholding the score of a predictive model as a decision rule influences the fairness of the final decisions. In particular, the authors consider binary decisions or predictions of a true binary outcome in presence of a sensitive binary attribute.  In that context, the authors show that the fairness measure of the total variation depends on the direct, indirect and spurious effects due to the true outcome and due the margin complement; a measure introduced by the authors expressing the difference between the predicted score and the final prediction after thresholding.  Under this new measure, the authors also introduce the notions of weak and strong business necessity (BN); under weak BN, disparities due to the true outcome are considered fair, while disparities due to the thresholding rule are not tolerated—zero marginal complement; under weak BN disparities due to both the outcome and the thresholding rule are tolerated.

**Strengths:**

The paper is well organized, has a clear structure and seem appropriately placed into contemporary literature.

The technical contributions appear strong, well motivated and clearly presented.  The examples do help the reader to understand the intuition and follow the technical results.

The problem of potential bias amplification due to applying a thresholding rule on predicted scores appears quite interesting and seems directly relevant and applicable in high stakes domains as the authors demonstrate through the real world data examples.

**Weaknesses:**

The abstract appears somewhat convoluted and confusing providing low level technical details, especially in lines 12-21. It might have been more helpful to explain the contribution at a higher level, while focusing more on the intuition.

**Questions:**

The causal decomposition of the TV measure are on the optimal 0/1 predictor. How this decomposition would change for a sub-optimal (e.g. near optimal) predictor?

**Limitations:**

The authors very briefly refer to the limitations due to the assumptions on the standard fairness model. It would be useful perhaps to elaborate concretely on some of these assumptions; e.g., on how the notion of the margin complement  could be extended assuming that the protected attribute takes values in a discrete set or a continuous range.

---

> ### Author Rebuttal · Authors · 2024-08-07
>
> Authors: We thank the reviewer for the time and effort in reviewing our paper. We are quite glad that the reviewer appreciated the ideas appearing in the paper, and considered the technical contributions strong, well motivated, and clearly presented. Below we address the main questions/concerns.
>
> (W1: Abstract) Thank you for this suggestion. Your point is indeed valid, it may be quite difficult to understand some of the technical detail that is mentioned in the abstract. This is now updated, and we provide a more high-level explanation of the developments.
>
> (Q1: Suboptimal predictors) This is a great question, and we thank the reviewer for raising this. In fact, this leads to an important result, which we add to the appendix (see also (P1) of the general response).
>
> Firstly, we note that Theorem 1 holds for any thresholded predictor $\tilde Y$ based on a score $\tilde S$. In other words, Theorem 1 is not true just for the optimal 0/1 predictor.
> Crucially, however, Theorem 2 no longer holds for a suboptimal predictor. We can clearly see that if $\tilde S$ is suboptimal, $E[\tilde S] = E[Y]$ is not expected to hold, which means that the causal effects of $X$ on $\tilde S$ need not equal those of $X$ on $Y$.
>
> Having said that, the suboptimality of $\tilde S$ can be remedied in a very nice way. In Corollary 3, we have a two-way decomposition along each causal path with contributions: (i) coming from the true $Y$; (ii) coming from the margin complement $M$. When instead of the optimal $S$, we are thresholding a suboptimal $\tilde S$, this results in three-way decomposition along each path with contributions: (i) coming from the original $Y$, (ii) contribution of the suboptimality (difference along the causal path resulting from using $\tilde S$ instead of $S$), and (iii) contribution of $M$. We invite the reviewer to check this interesting result in (P1) of the main review response. Thanks once again for pointing us in the right direction!

---

> > ### Comment · Reviewer_1zLn · 2024-08-13
> >
> > I would like to thank the authors for their reply that has sufficiently addressed my question. The authors should consider adding the discussion and results in (P1) in the revised version of their paper.

---

### Author Rebuttal · Authors · 2024-08-07

We thank the reviewers for their thoughtful reviews. We would like to mention three exciting updates to the paper that come as a result of some great questions from the reviewers. We believe these updates substantially improve the scope of the tools described in the paper:

(P1) (Reviewer 1zLn, Q1) We now introduce a new causal decomposition, which is a variation of Theorem 1 and Corollary 3. This decomposition handles threshold predictors that are based on arbitrary prediction scores $\tilde S$ (and not just the optimal predictor). The new decomposition is as follows:

$$
\\begin{align}
    \\text{TV}_{x_0, x_1}(\\tilde y) &=  {x\\text{-DE}\_{x_0, x_1}(y\\mid x_0)} +
 (x\\text{-DE}\_{x_0, x_1}(\\tilde s\\mid x_0) - x\\text{-DE}\_{x_0, x_1}(s\\mid x_0)) + {x\\text{-DE}\_{x_0, x_1}(\\tilde m\\mid x_0)}
         \\\\ &\\quad
        - \\big( {x\\text{-IE}\_{x_1, x_0}(y\\mid x_0)} + (x\\text{-IE}\_{x_1, x_0}(\\tilde s \\mid x_0) - x\\text{-IE}\_{x_1, x_0}(s\\mid x_0)) + {x\\text{-IE}\_{x_1, x_0}(\\tilde m\\mid x_0)} \\big)\\\\
        &\\quad- \\big( {x\\text{-SE}\_{x_1, x_0}(y)} + ({x\\text{-SE}\_{x_1, x_0}(s)} - {x\\text{-SE}\_{x_1, x_0}(\\tilde s)}) + {x\\text{-SE}\_{x_1, x_0}(m)} \\big).
\\end{align}
$$

As the expression indicates, there is an explicit term measuring how much the suboptimality of the prediction score $\tilde S$ contributes to the overall difference along the pathway. This is quantified by comparing the difference of the effect of $X$ on $\tilde S$ (the predictor being analyzed) vs. the effect of $X$ on $S$ (the optimal predictor).

(P2) (Reviewer ZV8n, W3) The reviewer raised the question of handling post-processing methods that use separate thresholds for different groups. As it turns out, our tools can be adapted to this setting as well. Suppose that $\tilde Y$ is a predictor using group-specific thresholds. In particular, a new TV decomposition can be written as follows:

$$
\\begin{align}
        \\text{TV}\_{x_0, x_1}(\\tilde y) &= x\\text{-DE}^{\\mathrm{GST}}\_{x_0, x_1}(\\tilde y \\mid x_0) +  {x\\text{-DE}\_{x_0, x_1}(\\tilde s\\mid x_0)} + {x\\text{-DE}\_{x_0, x_1}(\\tilde m\\mid x_0)}
        \\\\ &\\quad
        - \\big( {x\\text{-IE}\_{x_1, x_0}(\\tilde s\\mid x_0)} + {x\\text{-IE}\_{x_1, x_0}(\\tilde m\\mid x_0)} \\big)\\\\
        &\\quad- \\big( {x\\text{-SE}\_{x_1, x_0}(\\tilde s)} + {x\\text{-SE}\_{x_1, x_0}(\\tilde m)} \\big).
 \\end{align}
$$


In this new decomposition, we see that group-specific thresholds change only the direct effect of $X$ on $\tilde Y$ while the indirect and spurious effects remain the same. The term $x\text{-DE}^{\mathrm{GST}}_{x_0, x_1}(\tilde y \mid x_0)$ is given by

$$
\\begin{align}
E [ \\mathbb{1} ( \\tilde S_{x_1, W_{x_0}} (u) \\geq t_{x_1} ) - \\mathbb{1} ( \tilde S_{x_1, W_{x_0}} (u) \\geq t_{x_0} ) \\mid X = x_0 ]
\\end{align}
$$
and measures the direct effect of using a group-specific threshold. With this new decomposition, a whole class of methods in the fair ML literature can be analyzed, including reject-option classification (Kamiran & Calders, 2012) and post-processing for equality of odds (Hardt et. al., 2016), among many other methods.

(P3) (Reviewer bCLY, Q3 & Bounded Strong BN) The reviewer asked about different possible use cases for the notions of business necessity in Definition 3. Regarding the intended use cases, we provide the following explanations:

(i) No BN is used for pathways that are considered discriminatory, and when we wish to have no causal effect transmitted along the pathway.

(ii) Weak BN is used for cases where a pathway is important for the utility of the predictor designer, while there is still a need to avoid any kind of bias amplification compared to the current world.

(iii) Strong BN is intended for cases where the utility of the decision-maker along the causal pathway is so important that we are willing to accept even amplification of disparities as a result of improving this utility.

We remark that something in between Weak and Strong BN can be proposed, such as Bounded-Strong BN, where the contribution of the margin complement cannot be arbitrary but has to be bounded by some value $\alpha$. Therefore, we introduce the concept of $\alpha$-bounded Strong BN into the manuscript, which is discussed after Definition 3. We thank the reviewer for raising this question!

---

### Decision · Program_Chairs · 2024-09-25

**Decision:**

Accept (poster)

**Comment:**

The reviewers agreed that the paper studies an important problem of how thresholding prediction scores would influence the fairness of final decisions, and that the results would be of broad interest to the community. However, the reviewers also raised several concerns and questions in their initial reviews. We want to thank the authors for their responses and active engagement during the discussion phase. The reviewers appreciated the responses, which helped in answering their key questions. The reviewers have an overall positive assessment of the paper, and there is a consensus for acceptance. The reviewers have provided detailed feedback, and we strongly encourage the authors to incorporate this feedback when preparing the final version of the paper.